# A Technique for Monitoring Mechanically Ventilated Patient Lung Conditions

**DOI:** 10.3390/diagnostics14232616

**Published:** 2024-11-21

**Authors:** Pieter Marx, Henri Marais

**Affiliations:** Faculty of Engineering, North-West University, Potchefstroom 2531, South Africa; henri.marais@nwu.ac.za

**Keywords:** classification models, condition monitoring, lung, machine learning, mechanical ventilation, pulmonary diseases, regression models

## Abstract

Background: Mechanical ventilation is a critical but resource-intensive treatment. Automated tools are common in screening diagnostics, whereas real-time, continuous trend analysis in mechanical ventilation remains rare. Current techniques for monitoring lung conditions are often invasive, lack accuracy, and fail to isolate respiratory resistance—making them impractical for continuous monitoring and diagnosis. To address this challenge, we propose an automated, non-invasive condition monitoring method to support pulmonologists. Methods: Our method leverages ventilation waveform time-series data in controlled modes to monitor lung conditions automatically and non-invasively on a breath-by-breath basis while accurately isolating respiratory resistance. Results: Using statistical classification and regression models, the approach achieves 99.1% accuracy for ventilation mode classification, 97.5% accuracy for feature extraction, and 99.0% for predicting mechanical lung parameters. The models are both computationally efficient (720 K predictions per second per core) and lightweight (24.5 MB). Conclusions: By storing breath-by-breath predictions, pulmonologists can access a high-resolution trend of lung conditions, gaining clear insights into sudden changes without speculation and streamlining diagnosis and decision-making. The deployment of this solution could expand domain knowledge, enhance the understanding of patient conditions, and enable real-time dashboards for parallel monitoring, helping to prioritize patients and optimize resource use, which is especially valuable during pandemics.

## 1. Introduction

Mechanical ventilation (MV) is a critical intervention for patients with severe pulmonary diseases, providing essential respiratory support [1,2,3]. However, the administration of MV is resource-intensive, requiring specialized facilities and skilled healthcare practitioners. These resources are finite and can quickly become overwhelmed in high-demand situations, such as during pandemics or in regions with constrained healthcare infrastructure. According to the World Health Organisation’s 2019 global statistics, 35% of disease-related deaths target the pulmonary system [4]. This number has undoubtedly increased since then due to the COVID-19 pandemic [5].

Moreover, resource depletion extends beyond facilities; healthcare practitioners can also become overwhelmed, increasing the risk of potential errors. Such errors may involve infrequent patient assessments, inaccurate calculations, or inadequate record-keeping. Such lapses can result in suboptimal treatment, as practitioners may not have an accurate understanding of the patient’s condition, leading to improper patient prioritization and potentially adverse outcomes.

Previous research has tackled the challenges of mechanical ventilation by developing automated diagnostic methods aimed at improving condition monitoring for ventilated patients. These studies focused on estimating key respiratory parameters, such as resistance (*R*) and compliance (*C*), using mechanical ventilation waveform data, including airway pressure, flow, and volume time-series.

Many studies employed statistical methods, such as expiratory time constant calculations [6] or least squares fitting based on the single-compartment model’s motion equation [7,8,9,10,11,12,13]. Others utilized numerical techniques, including constrained optimization [14,15], iterative methods [16,17,18], and dynamic signal analysis [19,20]. Model-based methods were also applied [10,21,22,23]. One study in 2018 introduced a machine learning approach, albeit limited to the qualitative classification of respiratory mechanics as either low or high [24].

All these methods generally rely on validation against the end-inspiratory hold maneuver (EIHM), a classic manual technique for measuring passive respiratory mechanics [25]. EIHM, however, is resource-intensive, invasive, and disrupts the specific instance data, as it requires stopping airflow at end-inspiration to measure flow, pressure drop, and volume change. These measurements are subsequently used to estimate static compliance ($C_{S}$) and inspiratory resistance ($R_{INSP}$). A significant limitation of this technique is the frequent misinterpretation of $R_{INSP}$ as respiratory system resistance ($R_{RS}$), resulting in inaccuracies in previous studies’ models, which often treat these parameters as equivalent.

The accurate determination of target mechanical parameters presents additional challenges:In ICU settings, an endotracheal tube (ETT) bypasses a significant portion of the respiratory system, neglecting part of the true respiratory resistance.Most ventilators measure pressure at the mouth rather than the tracheal carina, leading to distortions from the ETT’s non-linear, flow-varying resistive component [25,26].The most accurate methods for obtaining these parameters involve either post-mortem analysis or transitioning patients to specialized equipment (e.g., forced oscillation techniques using Fourier transforms [27,28,29]), which are impractical and potentially hazardous.

Consequently, previous approaches have relied on flawed target values or required invasive and impractical procedures.

This study introduces an automated diagnostic technique for fully sedated, mechanically ventilated patients that is practical, non-invasive, mode-independent, and capable of tracking condition trends on a breath-by-breath basis. This approach offers multiple novel contributions. The approach relies solely on standard time-series data from the proximal airway (commonly measured at the mouth). It automatically classifies the ventilation mode in use, extracts relevant features and applies machine learning regression models to accurately quantify $C_{S}$ and $R_{RS}$, addressing the limitations of prior methods that relied on $R_{INSP}$ measurements. This technique not only has the potential to enhance patient diagnosis, prioritization, and treatment through precise monitoring and analysis, but it also alleviates healthcare providers’ workload, reduces resource demands, and minimizes the risk of manual errors in high-pressure scenarios.

## 2. Materials

Mechanically ventilated patients’ conditions are monitored in two primary ways: chemically driven (A-a gradient) [30,31,32] and data-driven (waveform data) [1,3,33]. The latter method is less invasive, more prompt, and simpler, involving the analysis of pressure, flow, and volume time-series (TS) data measured in the ventilatory circuit [30,34].

Since mechanical ventilation (MV) is a more appropriate treatment for ICU cases and carries risks, only modes synonymous with intubated patients are initially considered [35,36]. These modes include Volume-Controlled with Constant flow (VCC), Volume-Controlled with decreasing flow (VCD), and Pressure-Controlled (PC) modes [30,37,38,39,40,41]. Figure 1 shows that the VC modes aim to control the flow/volume waveform using ventilator parameters such as tidal volume ($V_{T}$) and maximum flow ($Q_{max}$), whereas PC aims to control the pressure waveform using the inspiratory time ($T_{i}$) and peak inspiratory pressure (PIP) [30,37,38,39,40,41]. These waveforms are considered the respective mode’s independent waveform. The ventilator administers this controlled ventilation waveform, the air interacts with the patient connected to the ventilator, and sensors measure the resulting dependent waveform [37,42]. The dependent waveforms are then analyzed to calculate the patient’s respiratory system resistance $R_{RS}$ and static compliance $C_{S}$, which describe the patient’s pulmonary health condition [1,3,33]. Note, that the inspiratory phase is an active process, whereas the expiratory phase is passive, during which the ventilator simply opens the expiratory valve, allowing equilibrium to occur based on the mechanical properties of the lungs and the expiratory circuit.

As illustrated in Figure 2, determining the trend of a patient’s condition is a non-trivial task that requires quantitative analysis and meticulous record-keeping. For example, note the waveform similarity of a healthy patient ($R_{RS}$: 8 ${\mathrm{cmH}}_{2}O\cdot s/L$ and $C_{S}$: 55 $L/{\mathrm{cmH}}_{2}O$) and an ill patient ($R_{RS}$: 15 ${\mathrm{cmH}}_{2}O\cdot s/L$ and $C_{S}$: 80 $L/{\mathrm{cmH}}_{2}O$) [37,38,43,44,45,46,47]. Without these calculations and detailed records, it becomes challenging to monitor patient progress and make informed clinical decisions.

To optimize resource use and improve patient recovery rates, it is essential to implement frequent and precise condition monitoring. This practice enables better patient prioritization. However, manual monitoring is not feasible due to the increased burden on practitioners, which exacerbates the current resource limitations. Therefore, automating the monitoring process is crucial. Automation not only frees up practitioners’ time for more critical tasks but also provides a scalable digital solution that can handle large volumes of data efficiently.

### 2.1. Automated Diagnostic Techniques in the Medical Domain

Automated diagnostic tools are widely used in medicine, demonstrating practitioners’ readiness to adopt technologies that enhance patient care.

Machine learning has been applied effectively in image processing for medical imaging (e.g., X-rays, CT scans, MRIs, mammograms) to aid in diagnosing conditions [48]. Statistical methods aid biochemistry analysis in automated blood analysis, proteomics, and genetic sequencing [49]. Additionally, time-series (TS) analysis techniques monitor vital signs such as EEG, ECG, blood pressure, and blood glucose levels [50]. These examples illustrate the broad potential for similar approaches in mechanical ventilation monitoring.

Most existing MV diagnostic systems focus on detecting patient–ventilator asynchrony (PVA) and adjusting settings to improve comfort [51,52]. In contrast, this study targets fully sedated patients, where PVA is not relevant. Our goal is to estimate key mechanical lung parameters, specifically respiratory system resistance ($R_{RS}$) and static compliance ($C_{S}$), which clinicians use to assess pulmonary health. Unlike PVA-focused systems, our work aims at diagnostic precision rather than real-time ventilator adjustments.

### 2.2. Time-Series Data Diagnostic Techniques in Medical Applications

Time-series (TS) diagnostic techniques are essential for analyzing dynamic medical data. These methods, applied across many domains, are broadly categorized into model-based, data-driven, and statistical approaches [53,54].

Model-Based Techniques: Techniques like Autoregressive Integrated Moving Average (ARIMA) and Kalman Filters help identify trends and patterns within physiological TS data by modeling the system’s behavior over time [55]. State Space Models and Hidden Markov Models further represent biological processes by tracking the changing states of these systems [56].

Data-Driven Techniques: Methods such as Fourier [57] and Wavelet Transforms [58] detect periodic patterns and abrupt shifts, while machine learning models like Long Short-Term Memory (LSTM) networks [59] and Convolutional Neural Networks (CNN) [60] analyze complex TS data to predict critical events. Dynamic Time Warping (DTW) and clustering allow the comparison and classification of physiological phases, supporting early detection of health changes [61]. Principal Component Analysis (PCA) reduces data dimensionality, emphasizing significant factors for analysis [62].

Statistical Techniques: Techniques like anomaly detection identify outliers, a key step for early diagnosis, while Granger Causality identifies potential causal links among health variables [63].

In medical applications, a preliminary step in using these techniques is feature engineering, where significant attributes are extracted from TS data to enhance diagnostic accuracy [64]. Despite their success in many fields, applying TS diagnostic techniques to MV monitoring is challenging due to patient variability, limited data quality, and real-time data access constraints.

### 2.3. Challenges in Applying Time-Series Diagnostic Techniques to MV Data

MV patient monitoring presents unique challenges that limit the direct application of established TS diagnostic techniques. For effective automated monitoring, high-quality, diverse datasets are essential, but in MV, data scarcity and quality issues hinder diagnostic accuracy.

Existing MV datasets often suffer from the following:Inappropriate Data Types: Many datasets focus on limited patient condition labels or data from non-ideal ventilation modes like BiPAP/CPAP.Irrelevance to Fully Sedated Patients: Data related to PVA are irrelevant here, as our focus is on fully sedated patients.Incompleteness and Low Sampling Frequency: Many datasets have infrequent data collection, reducing their utility for accurate diagnostics.Limited Data Volume and Variety: Available data lack diversity in patient parameters and are not representative of broader populations.Lack of Labeling: Many datasets lack essential information such as ventilator settings and patient characteristics.

These issues stem partly from privacy restrictions, data collection complexities, and clinical variations. Addressing these limitations is crucial to advance automated MV monitoring.

### 2.4. Synthetic Data and Human Lung Modeling for MV Diagnostics

Due to MV data scarcity, synthetic data offer an alternative. Two main types of synthetic data are used in MV research: emulated data, which replicate physical processes, and simulated data, which are generated through mathematical models.

Lung models in the literature vary in complexity:Single-Compartment Models: RC, RIC, and extended RIC models represent basic resistance and compliance [38,65].Multi-Compartment Models: Series RC, parallel RC, and bi-compartmental models [66].Advanced Models: Models like Mead, Dubois, and viscoelastic models incorporate complex mechanical behaviors [65].

This study focuses on estimating the trend of the MV patient’s $R_{RS}$ and $C_{S}$, as these are the only two mechanical parameters practitioners use for monitoring a patient’s pulmonary health. The inertial component is typically negligible in clinical considerations [43,65,67]. Among these human lung models, the single-compartment model is preferred by practitioners for its simplicity and sufficient representative complexity in deriving respiratory resistance and static compliance from its data, rendering the other complex models unnecessary. Therefore, it is advised to compare existing single-compartment RC simulation models and select the most suitable one to generate the data required for this study.

### 2.5. Current Mechanical Parameter Analysis Techniques

Estimating lung parameters ($R_{RS}$ and $C_{S}$) remains challenging.

The primary technique, the end-inspiratory hold maneuver (EIHM) [68], temporarily halts airflow to stabilize airway pressure, enabling calculations of static compliance ($C_{S}$) and inspiratory resistance ($R_{INSP}$), which is often used as an approximation for $R_{RS}$. This makes EIHM impractical for continuous, breath-by-breath monitoring. Additionally, approximating $R_{RS}$ using $R_{INSP}$ is limited by the added resistance from the endotracheal tube (ETT), which varies non-linearly with flow rates and can be affected by turbulent airflow conditions. With 17 standard ETT sizes [69,70], isolating the ETT resistance component from $R_{INSP}$ is challenging, and no current automated technique effectively extracts $R_{RS}$ from $R_{INSP}$. This is problematic since $R_{INSP}$ can exceed $R_{RS}$ by more than twofold in some cases [65].

Since EIHM is not feasible for continuous monitoring, studies on automated techniques (mentioned in Section 1) commonly estimate $C_{S}$ with dynamic compliance ($C_{D}$) [1,3,33]. However, this estimator’s accuracy fluctuates significantly with changes in $R_{INSP}$, which are unknown without EIHM.

Flow-volume loops provide another approach to qualitatively assess $R_{RS}$ and $C_{S}$. The loop’s expiratory phase provides useful visual cues: its concavity correlates with $R_{RS}$ (Figure 3a), while its gradient is inversely proportional to $C_{S}$ (Figure 3b) [68]. Quantitative methods linking these visual patterns to precise values for $R_{RS}$ and $C_{S}$ remain undeveloped.

## 3. Methods

### 3.1. Synthetic Data Generation

To address the scarcity of suitable mechanical ventilation (MV) datasets, we utilized synthetic data generated through simulation. Among the available models [45,71,72,73,74,75,76,77,78,79,80,81,82,83], we identified one that is most appropriate for our purposes and is a validated and verified patient–ventilator data generating tool published in [71]. The tool was validated by ensuring the correct generation of independent waveforms in accordance with the set ventilator parameters and by confirming that the resulting dependent waveform responded appropriately to changes in the patient’s pulmonary health parameters. Validation involved overlaying the simulated time-series (TS) waveform data with real-world data for comparison.

Although this tool allows for extensive simulation scenarios, the source also published a generated dataset comprising 640 K TS breaths (pressure, flow, and volume over time) for various ventilation modes (VCC, VCD, and PC) along with their respective mode parameters, equating to 1.92 M unique breaths.

The model parameters included the following:VCC and VCD Modes: Positive End-Expiratory Pressure (PEEP), Maximum Flow ($Q_{max}$), Tidal Volume ($V_{t}$), Respiratory Resistance ($R_{RS}$), Static Compliance ($C_{S}$)PC Mode: PEEP, Peak Inspiratory Pressure (PIP), Inspiratory Time ($T_{i}$), $R_{RS}$, $C_{S}$

The dataset was designed to encompass a diversity of realistic scenarios by sweeping the ventilator settings and patient parameters across the following ranges:PEEP: 0 to 15 ${\mathrm{cmH}}_{2}O$ (4 levels)$Q_{max}$: 10 to 105 L/min (20 levels)$V_{t}$: 180 to 750 mL (20 levels)PIP: PEEP + 5 to 35 ${\mathrm{cmH}}_{2}O$ (20 levels)$T_{i}$: 0.5 to 2.4 s (20 levels)$R_{RS}$: 1 to 20 ${\mathrm{cmH}}_{2}O\cdot s/L$ (20 levels)$C_{S}$: 10 to 105 $L/{\mathrm{cmH}}_{2}O$ (20 levels)

### 3.2. Automated Diagnostic Approach

The proposed automated diagnostic approach comprises three main sequential components (see Figure 4). First, the automated statistical mode classifier analyzes time-series data to identify the administered ventilation mode. The classifier’s output informs both the selection of appropriate regression models and the extraction of mode-dependent (patient-independent) features within the feature extraction layer, which also processes time-series data. These extracted features, combined with the ventilation mode feature, form the feature vector, which serves as the input for the selected machine learning regression model to estimate mechanical lung parameters. The mode classifier was trained on a representative dataset of 30,000 breaths (10,000 per mode) from a total dataset of 1.92 million breaths, whilst the regression models were trained on 80% of the complete dataset.

#### 3.2.1. Automatic Mode Classifier

Given the variation in how different ventilation modes respond to the same conditions and adjustments, it is essential to classify the ventilation mode as a feature before inputting TS and anthropometric data into machine learning models. We developed an automatic mode classifier using statistical techniques to analyze TS waveforms and determine the ventilation mode in use. The classifier leverages statistical parameters as informative features that describe the waveform shapes during the inspiratory phase.

The key parameters used for comprehensive classification include the following:Standard Deviation (STD) from the Average Inspiratory Pressure: Represented by a horizontal line during the inspiratory phase of a normalized waveform. It is small for PC mode, slightly larger for VCD mode, and largest for VCC mode.Coefficient of Determination ($R^{2}$) of the Flow Waveform: Measures the fit of the flow waveform to a straight line spanning from the point of maximum initial flow to zero flow at the end of the inspiratory phase. It indicates a good fit for VCD mode, a fair fit for PC mode, and a poor fit for VCC mode.

Figure 5 illustrates the concept for overlaying the reference lines (in red) over the waveform data, as described above, in order to extract these ventilation mode description features from the inspiratory phase.

These descriptive features were extracted from the training dataset, with their ranges shown in the boxplots in Figure 6.

Initially, a three-layer decision tree approach was manually developed by visually and iteratively identifying hard decision boundaries on the boxplots of these extracted parameters, classifying up to 92.36% of all instances. The remaining 7.64% (see Figure 7) is due to the close resemblance (confusion) between VCD and PC. Assigning the remaining instances to the mode with predominant unclassified instances (PC mode) results in a 96.42% true positive rating.

Another approach would be to instead use kernel probability distribution functions (KPDFs [84]) to map the likelihood distribution of these statistical waveform shape features (see Figure 8). Solely applying KPDFs results in only a 94.24% true positive rate.

However, we can use the KPDFs approach on the remaining unclassified instances from the aforementioned decision tree approach (see Figure 9). This hybrid classifier results in a 99.18% true positive rate on the training dataset.

#### 3.2.2. Feature Extraction Layer

The feature extraction layer focuses on deriving meaningful features from patient-independent and patient-dependent TS waveform data for use in regression models.

Patient-Independent Features:Vent Mode: VCC, VCD or PCVent Parameters: PEEP, $Q_{max}$, $V_{t}$, PIP, $T_{i}$

These patient-independent features were chosen because they provide a description of the shape of the waveform, which is critical for contextualizing the administered ventilation protocol.

Patient-Dependent Features:Expiratory Phase Volume Time Constant ($\tau_{RC}$ in s)Expiratory Flow at One Time Constant ($Q_{\tau }$ in L/min)Flow-Volume Loop Shape Descriptors: gradient, surface ratio, skewness, kurtosis

These patient-dependent features were chosen because they provide insights into the shape and dynamics of the expiratory phase (mode-independent and patient condition-dependent phase) of the flow-volume loop, which are critical for deriving the mechanical properties of the lung.

Figure 10 clarifies the meaning of the flow-volume loop scoop shape descriptors. The gradient is simply the maximum negative flow divided by the maximum volume displacement. The surface ratio is the area encompassed by the actual scoop (grey and blue shaded area) divided by the area encompassed by the gradient line (blue triangular area).

Skewness [85] and kurtosis [86] further characterize the curve’s shape (blue line) during the exhalation phase by quantifying the horizontal and vertical asymmetry of the data-point distribution (blue squares) relative to the mean. However, the data points were sampled at 200 Hz (time-dependent). Therefore, the original data-point distribution is irregular when omitting the time dimension to plot the flow-volume loop (see Figure 11a) as an example). The skewness and kurtosis functions in MATLAB^®^ only accept the value array as input (flow values). Therefore, to conserve representative information, the points need to be resampled (interpolated) over the same curve concerning the parameter array (volume values), as illustrated by Figure 11b. After that, the skewness and kurtosis of the resampled flow data points can be extracted as features to further quantitatively describe the scoop shape.

#### 3.2.3. Supervised Regression Models for Lung Parameters

A supervised regression approach was used to predict $R_{RS}$ and $C_{S}$ based on the same training dataset from the mode classification stage and results from the feature extraction layer. The MATLAB Regression Learner Toolbox [87] was selected for its ease of use and automated processes, which allow for minimal user-defined settings. This approach aligned with the study’s goal of assessing model performance rather than optimizing it.

To ensure consistent training, we specified essential parameters: all extracted features were used as predictors, and each regression model was trained to predict one mechanical parameter (either $R_{RS}$ or $C_{S}$) as the response variable. For validation, a five-fold cross-validation scheme was chosen, with an additional 20% of the dataset set aside for independent testing. While additional settings, such as hyperparameters, training options, and random seed definitions, are available for further customization, we retained default values to maintain a baseline comparison across models.

The Regression Learner Toolbox trained 24 different regression algorithms across six scenarios (three modes and two mechanical parameters). The regression algorithms included the following:Linear Regression Algorithms: Normal Linear Regression, Interactions Linear Regression, Robust Linear Regression, and Stepwise Linear RegressionDecision Tree Algorithms: Fine Trees, Medium Trees, Coarse TreesSupport Vector Machine (SVM) Algorithms: Linear SVM, Quadratic SVM, Cubic SVM, Fine Gaussian SVM, Medium Gaussian SVM, and Coarse Gaussian SVMGaussian Process Regression (GPR) Algorithms: Exponential GPR, Squared Exponential GPR, Matern 5/2 GPR, Rational Quadratic GPREnsemble Algorithms: Boosted Trees, and Bagged TreesNeural Networks (NN): Narrow Neural Network (NNN), Medium Neural Network (MNN), Wide Neural Network (WNN), Bi-Layered Neural Network (BLNN), and Tri-Layered Neural Network (TLNN).

Performance metrics, including RMSE percentage, training speed, over-fitting index, prediction speed, and model size, are used to evaluate and compare the models.

## 4. Results

Following the development of the hybrid automatic mode classifier, feature extraction layer, and supervised regression models, the remaining 1.92 million breaths were used as the testing set, and the results were recorded. The predictions of the testing set were validated against the provided labeled data from the dataset source. These labels include the ventilation modes and the actual $R_{RS}$ and $C_{S}$ mechanical parameters, showcasing unmatched accuracies of over 98%.

### 4.1. Automatic Mode Classifier

During training, the true positive rates were 100.00% for VCC mode, 98.67% for VCD mode, and 98.86% for PC mode. In comparison, across the remaining 1.92 million breaths, the true positive rates slightly decreased to 99.87%, 98.75%, and 98.71%, respectively. The false positive rates for VCC, VCD, and PC modes were 0.03%, 1.26%, and 1.38%. Despite this, the classifier’s overall accuracy only dropped slightly, from 99.18% to 99.11%, when evaluated on a dataset that was 63 times larger, demonstrating that the proposed hybrid classifier generalizes well.

### 4.2. Feature Extraction Layer

The feature extraction layer effectively derived both independent and dependent features, as outlined in Section 3.2.2. Independent features, derived from ventilator settings, allow for the direct evaluation of accuracy and precision, unlike the dependent features. The average and standard deviation, summarized in Table 1, serve as indicators of the accuracy and precision of the feature extraction layer. The extracted $Q_{max}$ values are relatively lower than the other features due to the simulation model using flow-lagging mechanical valves, causing a lower inspiratory flow than what was set. Also, since $V_{T}$ is the integral of $Q_{max}$, the simulation error propagates through to this feature. Therefore, these inaccuracies are not due to the feature extraction layer, but rather a simulation tool problem. Nonetheless, the total feature extraction layer residual mean percentage is −2.423% and the residual standard deviation percentage is 3.514%.

### 4.3. Supervised Regression Models for Lung Parameters

After training 24 different algorithms for each of the six scenarios using the MATLAB Regression Learner Toolbox, the testing set was applied to each and the performance of each model was recorded.

Performance metrics, including RMSE percentage, training speed, overfitting index, prediction speed, and model size were used to evaluate and compare the models.

#### 4.3.1. RMSE Percentage

The root mean squared error of the predictions versus the target values (accuracy). The absolute percentage range is from 0.167 to 9.815% for $R_{RS}$ and from 0.244 to 14.921% for $C_{S}$. Figure 12 shows the boxplots of the RMSE percentage spread for the PC scenarios. The dots represent data points, the cross denotes the mean, the short lines indicate the interquartile range, and the long lines (whiskers) show the range of values within 1.5 times the interquartile range. The cut-off criteria for the RMSE percentage is chosen as 5.000%.

#### 4.3.2. Training Speed

The duration required to train a model using the specified algorithm. The absolute range is 0.000 to 20.844 days for $R_{RS}$ and 0.000 to 11.015 days for $C_{S}$. Figure 13 shows the boxplots of the training speed spread for the PC scenarios. The cut-off criteria for the training speed is chosen as 4.000 days. Since training is conducted infrequently, this performance indicator’s scoring weight is decreased ten-fold.

#### 4.3.3. Overfitting Index

An indication of the quality of generalization. It is derived from the validation and test RMSE results. The closer these RMSE results are, the smaller the likelihood that the model is overfitted. The absolute range is 0.000 to 287.940% for $R_{RS}$ and 0.085 to 167.087% for $C_{S}$. Figure 14 shows the boxplots of the overfitting index spread for the PC scenarios. The cut-off criteria for the overfitting index is chosen as 10.000%.

#### 4.3.4. Prediction Speed

The number of predictions a model can generate within a certain amount of time can greatly affect the deployment cost. The absolute range is 87 to 990,000 obstacles/s (Obs/s) for $R_{RS}$ and 90 to 1,100,000 Obs/s for $C_{S}$. Figure 15 shows the boxplots of the prediction speed spread for the PC scenarios. The cut-off criteria for the prediction speed is chosen as 300,000 Obs/s.

#### 4.3.5. Model Size

The storage space or memory required to host a model instance on a machine can significantly affect deployment costs. The absolute range is 23.103 to 238.944 MB for $R_{RS}$ and 23.204 to 239.063 MB for $C_{S}$. Figure 16 shows the boxplots of the model size spread for the PC scenarios. The cut-off criteria for the model size is chosen as 55.000 MB.

The cut-off criteria, along with adjusted bias weights (training time weight reduced tenfold, others set to unity), were applied to evaluate each model. The top-performing models for each scenario are summarized in Table 2.

Figure 17 presents boxplots of the residuals across the testing dataset for some PC mode models, illustrating (a) resistance (worst), (b) resistance (best), (c) compliance (worst) and (d) compliance (best). These plots demonstrate that both $R_{RS}$ and $C_{S}$ exhibit relatively consistent accuracy across the target range when using the best overall performing models: (b) for $R_{RS}$ and (d) for $C_{S}$. However, the residuals show increased variance or widening dispersion at the upper range, suggesting that prediction accuracy decreases as the true response values reach the upper limits.

Note that from Table 2, neural networks are considered the best-performing algorithms in the initial iteration of this study. Also, the worst accuracy is 97.703%. Training times for these models vary significantly, ranging from 0.67 to 81 h. The overfitting indices are consistently low, indicating strong generalization across the models. Prediction speeds were achieved between 0.33 and 1.1 M observations per second. Additionally, the model sizes average around 25 MB.

### 4.4. Deployment of Regression Models

The selected regression models for all modes were deployed and validated using simulated mechanical ventilator sessions with varying patient conditions ($R_{RS}$ or $C_{S}$). The simulations utilized a generic patient profile: a 25-year-old male, 186 cm in height. An example of PC mode deployment is provided, using baseline ventilator settings that include a pressure of 5 ${\mathrm{cmH}}_{2}O$, a peak inspiratory pressure of 24.5 ${\mathrm{cmH}}_{2}O$, and an inspiratory time of 2.0 s. The true response trend begins with the patient in a stable, mid-level condition, followed by an increase in the respective lung parameter until the maximum value is reached, after which the condition reverses, decreasing to the lower end of the spectrum. Time-series waveform data from the ventilator under these conditions are processed through the automatic hybrid mode classifier and feature extraction layer. A feature vector for each breath is then passed in parallel to the resistance and compliance regression models. These parameters are tracked over time to generate a condition trend, reflecting the patient’s respiratory state. Figure 18 presents a comparison between the predicted and true condition trends, demonstrating the close alignment between them and validating the research’s outcomes for (a) resistance and (b) compliance.

Table 3 summarizes the average RMSE percentages across all six trend simulation scenarios. The local RMSE percentage is calculated relative to the individual local target values, whereas the total RMSE percentage is based on the entire range of the target values. All six simulated use cases achieved accuracies exceeding 98.22% across the total range of target values.

## 5. Discussion

This paper’s proposed automatic condition monitoring method is the first to utilize waveform data from continuous mandatory ventilation modes, measured at the mouth opening, to accurately quantify actual respiratory resistance ($R_{RS}$) and static compliance ($C_{S}$) over a spectrum using machine learning techniques.

Effective feature engineering is a vital component of any machine learning experiment. This requires not only that the accuracy and precision of feature extraction meet acceptable standards, but also a thorough exploration of the data to identify correlations and patterns, which can then be distilled into relevant and comprehensive descriptive features. This motivated the development of the automatic hybrid mode classifier and feature extraction layer, which achieved acceptable average errors of 0.89% and −2.42%, respectively.

Table 4 presents a comparative overview of the performance characteristics of the different clusters of machine learning regression models. It is important to note that these results pertain to one use case and may vary for other scenarios and projects.

Neural networks demonstrated the highest overall performance during the initial phase of this study, achieving a minimum accuracy of 97.703%. This performance is exceptional, especially when compared to the worst-case accuracy of existing mechanical parameter analysis techniques, which may be overestimated by more than double (Section 2.5). The combination of compact model sizes (approximately 25 MB) and high prediction speeds (each model executed on a single Intel^®^ Core™ i7-6700HQ CPU core @ 2.60 GHz, manufactured by Intel Corporation, Santa Clara, CA, USA) results in minimal computing resources, facilitating scalability for monitoring multiple patients concurrently. For example, assuming similar core performance, resistance predictions for a breath with pre-processed feature values can theoretically be completed at a rate exceeding 610,000 breaths per second. Given that the average duration of a full respiratory cycle is over 3 s, a single core could theoretically monitor more than 1.83 M patients simultaneously. Thus, these models offer cost-effective and practical solutions to current challenges in pulmonary care.

An important consideration, despite this being an initial iteration, is that the regression models do not maintain constant accuracy across the entire target range, as illustrated in Figure 17. If one model generalizes well in the lower end of the spectrum and another performs better in the higher end, these models could potentially be concatenated. A pre-prediction model combined with a decision tree could then be employed to determine which model to apply for a given range, thereby leveraging the strengths of both models.

Once the models were developed and the feature vectors necessary for predicting a patient’s health condition for each breath were identified, a breath-by-breath condition monitoring system could be implemented to track the patient’s condition trends. These data would provide practitioners with significant advantages, not only by automating and recording calculations—thereby streamlining the screening process, saving time, enabling more frequent assessments, and maintaining data integrity—but also by providing a straightforward method to assess the direction and magnitude of changes in the patient’s condition. This enables more informed decisions regarding treatment modifications and patient prioritization.

Such a system could, for example, be valuable in detecting patient–ventilator asynchrony (PVA), such as double-triggering or breath stacking. During assisted mechanical ventilation, a patient can trigger the ventilator to deliver a breath outside of the inspiratory phase. This becomes problematic when a patient with highly compliant lungs attempts to take an additional breath during the exhalation phase, leading to lung overinflation and increased intrapulmonary pressure. This may result in sensor misreadings, indicating falsely low lung compliance, which, if undetected, could lead to misdiagnosis and inappropriate treatment. By recording the breath-by-breath trends, pulmonologists could more easily detect PVAs like double-triggering, thereby reducing the risk of mistreatment.

Although the lung model used in this study was homogeneous, the overall simulation incorporated additional components, such as the endotracheal tube (ETT), which introduced some heterogeneity. Our results demonstrated that with appropriately labeled data, the effects of individual components, like the ETT, could be isolated effectively, highlighting the potential for further model complexity. The authors anticipate that increasing the resolution of the simulation model to account for lung heterogeneity—such as differences between dependent and non-dependent lung zones—could enable machine learning to identify specific regional effects of pulmonary disease. This approach could isolate parameters at a higher resolution, potentially offering pulmonologists more precise diagnostic information. We, therefore, believe that a more complex model, simulating heterogeneous lung characteristics, could enhance the application of machine learning in pulmonary disease classification and improve diagnostic accuracy.

## 6. Work Limitations

One key limitation of the proposed models is their dependency on passive patient conditions; they may become inaccurate if a patient exerts any work of breathing. The models were specifically designed for use in the top three continuous mandatory ventilation (CMV) modes, limiting their applicability to other ventilation modes where patient-initiated breaths and patient–ventilator asynchrony are prevalent. Extending the models to account for active breathing efforts or additional ventilation modes would require additional research.

Another limitation is the narrow scope regarding endotracheal tube (ETT) sizes. While the proposed method successfully eliminates the non-linear resistance component associated with the ETT, the models were trained and validated on data from a single ETT size. To generalize the models for broader clinical applications, they must be expanded and validated across various ETT sizes to account for resistance variations caused by differing diameters and lengths.

The models may also be affected by differences in ventilator hardware. Different ventilators exhibit varying levels of accuracy and transient response characteristics (e.g., valve types and responsiveness), which could impact the models’ predictive accuracy. Additionally, not all ventilators provide access to the necessary waveform data, which may necessitate the development of a custom pressure and flow measurement device. Such a device must be carefully designed to avoid interfering with the ventilation process and to meet the ethical standards for patient safety.

Lastly, the issue of sensor noise presents a limitation in the model’s practical implementation. The waveform data obtained from sensors may contain noise, which could reduce model accuracy if not adequately filtered. As no sensitivity tests have been conducted, the models’ susceptibility to sensor noise or other data artifacts remains uncertain. Pre-processing techniques to clean the waveform data may be required to ensure reliable model performance in a clinical setting.

## 7. Conclusions

This study highlights the potential of automated machine learning models to address challenges in mechanical ventilation, particularly in resource-limited settings. By predicting key pulmonary parameters, such as respiratory resistance ($R_{RS}$) and static compliance ($C_{S}$), these models provide healthcare practitioners with real-time, accurate insights into patient conditions. This can reduce the burden on healthcare professionals while improving patient outcomes through more frequent and reliable monitoring.

Neural networks showed promising results, with exceptional accuracy, compact model sizes, and efficient prediction speeds, making them scalable for monitoring multiple patients. Some variability in performance across the target range suggests room for further optimization.

Monitoring trends in resistance and compliance over time could enhance diagnosis and treatment. These trends may not only help assess the severity of respiratory conditions but also support diagnostic decisions, potentially automating the process or providing intelligent suggestions, allowing practitioners to focus on other critical tasks.

A scalable dashboard connected to pressure, flow, and volume sensors could capture waveform data from thousands or millions of patients. Such a system would enable a non-invasive, breath-by-breath analysis of respiratory parameters, enhancing record-keeping, practitioner efficiency, and patient prioritization, while alleviating strain during respiratory pandemics. Access to real-time data would ultimately increase the likelihood of successful patient recoveries, potentially saving lives.

Additionally, collecting and analyzing data over time could deepen our understanding of respiratory disease progression and recovery, guiding future treatment strategies and enhancing long-term outcomes.

In conclusion, continued refinement and integration of these models with automated diagnostic tools and scalable infrastructure have the potential to revolutionize the monitoring, diagnosis, and treatment of pulmonary conditions, driving significant advancements in our understanding of respiratory diseases and global healthcare practices.

## Figures and Tables

**Figure 1 diagnostics-14-02616-f001:**
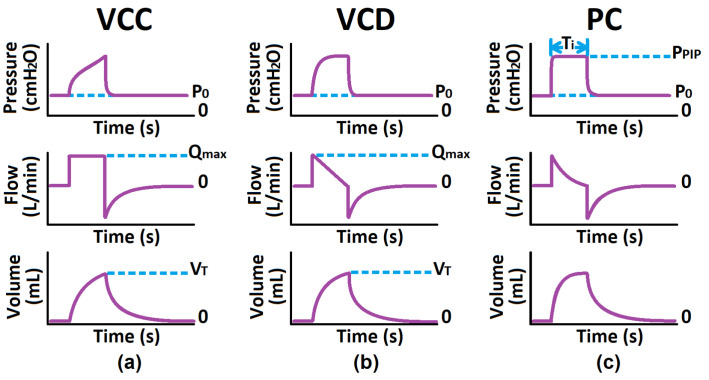
Common Critical, Mandatory Modes: (**a**) Volume-Controlled Constant Flow Pattern, (**b**) Volume-Controlled Decelerating Flow Pattern, (**c**) Pressure-Controlled.

**Figure 2 diagnostics-14-02616-f002:**
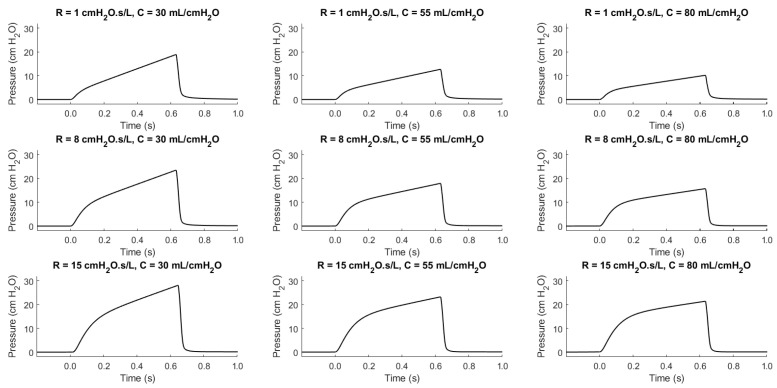
The pressure-dependent waveforms of the VCC mode for swept patient health conditions.

**Figure 3 diagnostics-14-02616-f003:**
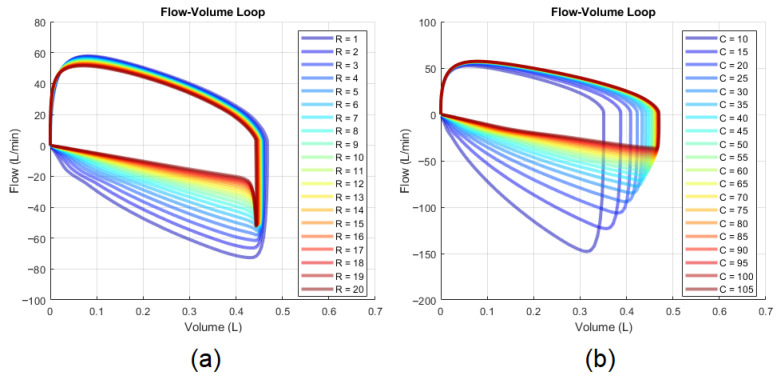
Effects on the flow-volume loop: (**a**) changes in $R_{RS}$ and (**b**) changes in $C_{S}$.

**Figure 4 diagnostics-14-02616-f004:**
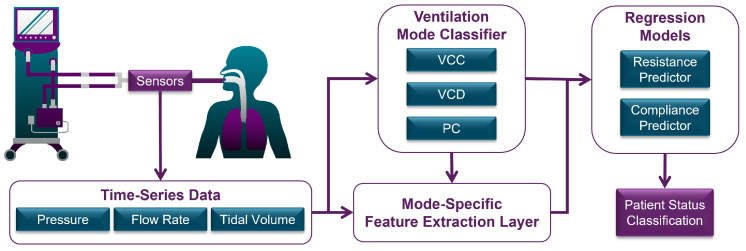
Flow diagram of total system overview.

**Figure 5 diagnostics-14-02616-f005:**
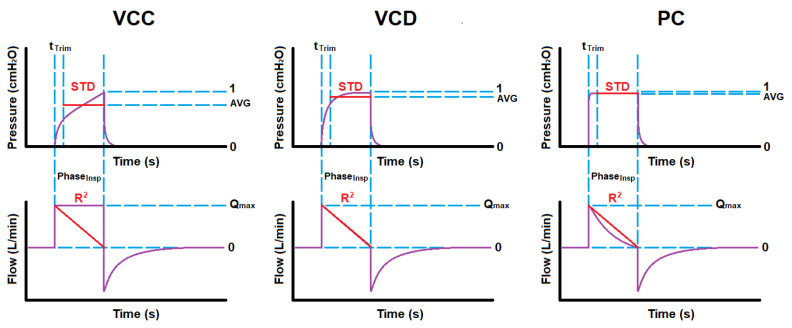
Typical reference overlay onto waveforms for extracting standard deviation and coefficient of determination per ventilation mode.

**Figure 6 diagnostics-14-02616-f006:**
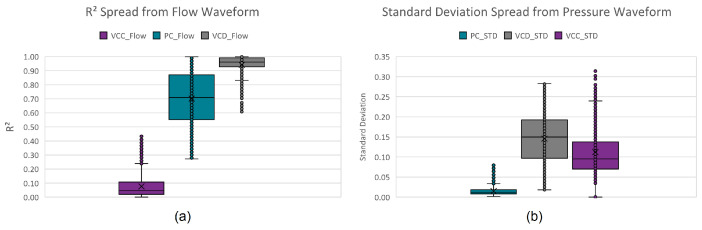
Per mode boxplots of the descriptive features: (**a**) standard deviation and (**b**) coefficient of determination.

**Figure 7 diagnostics-14-02616-f007:**
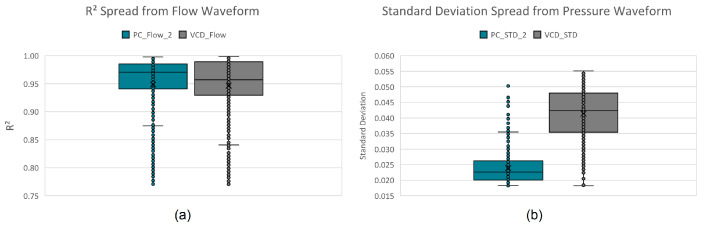
Per mode confusion boxplots of the descriptive features: (**a**) standard deviation and (**b**) coefficient of determination.

**Figure 8 diagnostics-14-02616-f008:**
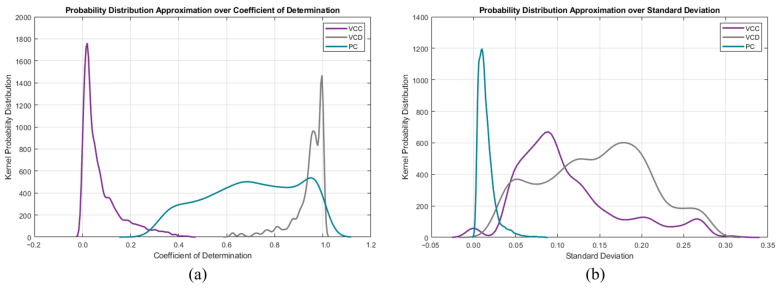
Per mode kernel probability distribution functions of the descriptive features: (**a**) standard deviation and (**b**) coefficient of determination.

**Figure 9 diagnostics-14-02616-f009:**
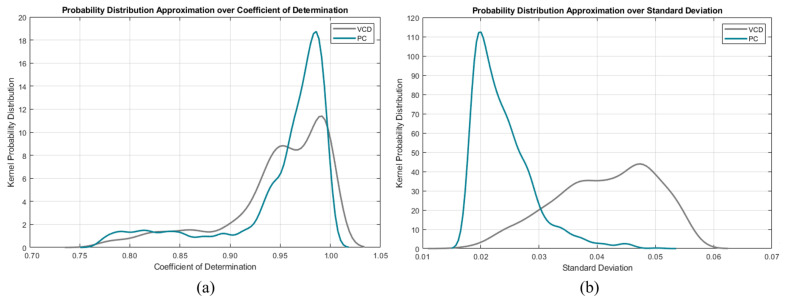
Per mode confusion kernel probability distribution functions of the descriptive features: (**a**) standard deviation and (**b**) coefficient of determination.

**Figure 10 diagnostics-14-02616-f010:**
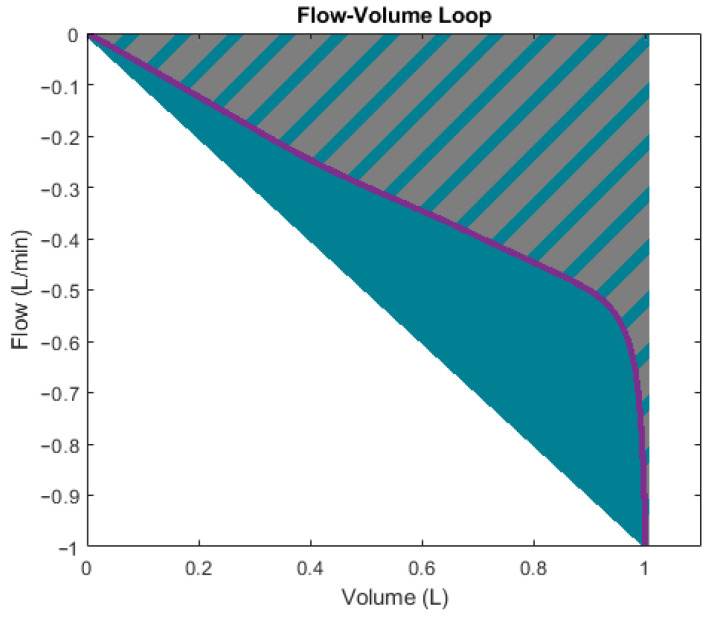
Expiratory phase of flow-volume loop with shape descriptors.

**Figure 11 diagnostics-14-02616-f011:**
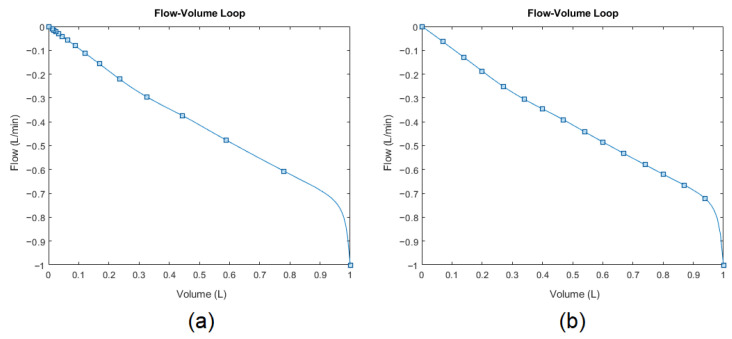
Expiratory phase of flow-volume loop: (**a**) originally sampled distribution and (**b**) resampled distribution for conserving representative information.

**Figure 12 diagnostics-14-02616-f012:**
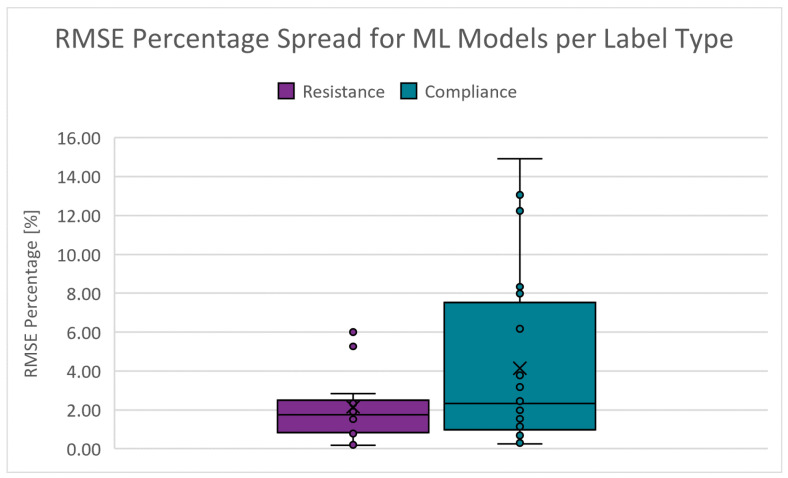
Boxplots of the RMSE percentage for PC scenarios.

**Figure 13 diagnostics-14-02616-f013:**
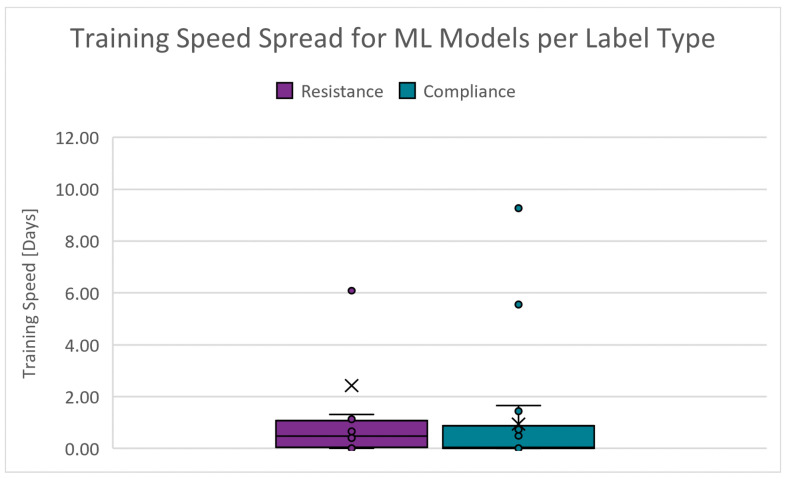
Boxplots of the training speed for PC scenarios.

**Figure 14 diagnostics-14-02616-f014:**
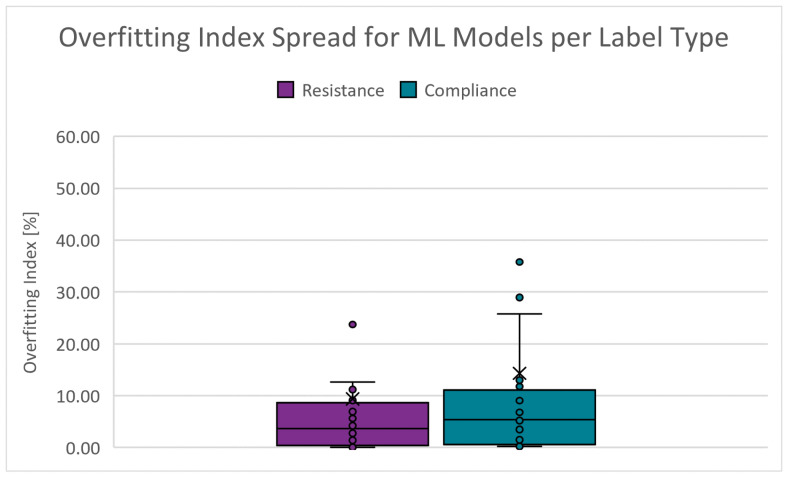
Boxplots of the overfitting index for PC scenarios.

**Figure 15 diagnostics-14-02616-f015:**
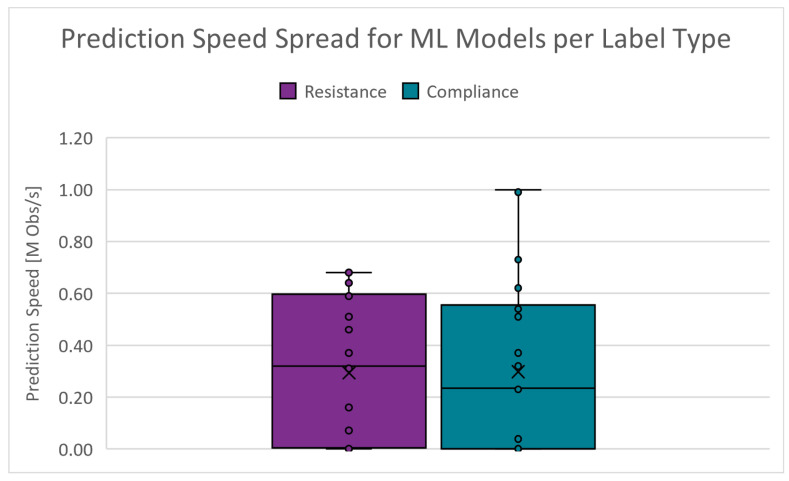
Boxplots of the prediction speed for PC scenarios.

**Figure 16 diagnostics-14-02616-f016:**
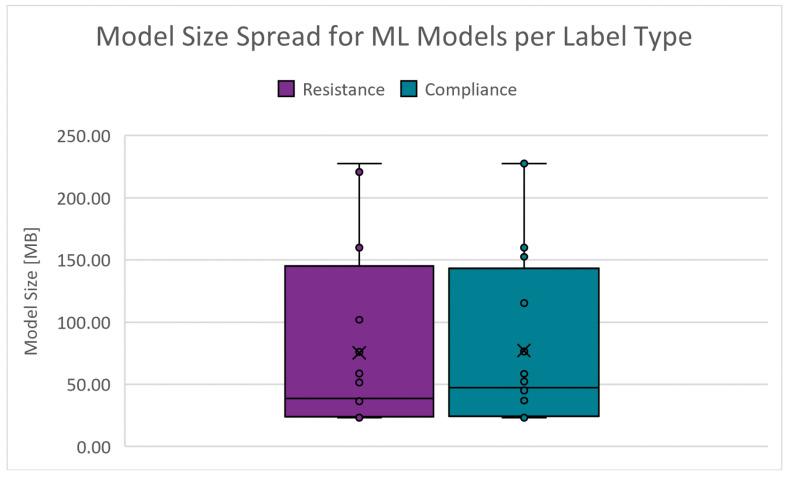
Boxplots of the model size for PC scenarios.

**Figure 17 diagnostics-14-02616-f017:**
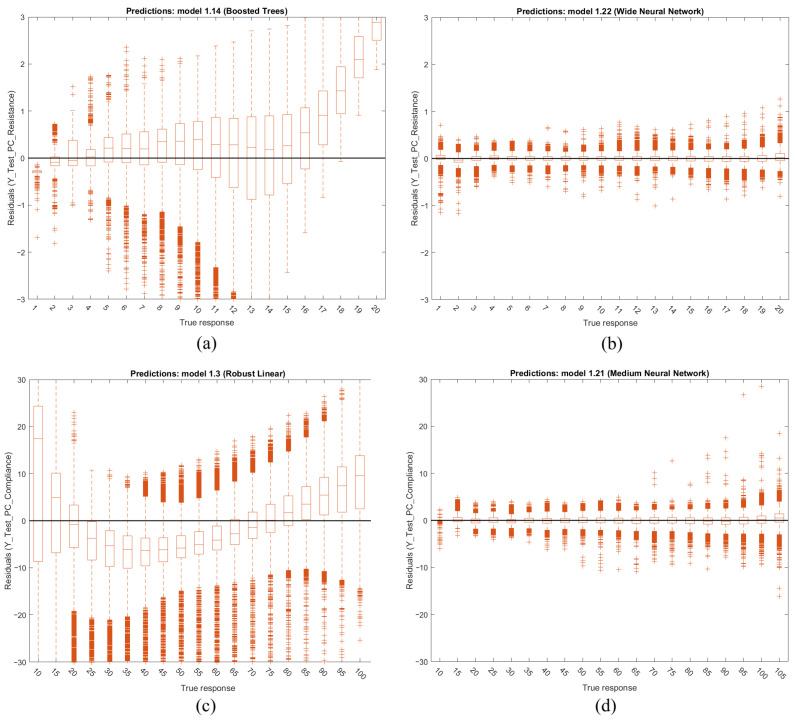
Boxplots of the testing datasets residuals of PC for (**a**) $R_{RS}$ (worst), (**b**) $R_{RS}$ (best), (**c**) $C_{S}$ (worst) and (**d**) $C_{S}$ (best).

**Figure 18 diagnostics-14-02616-f018:**
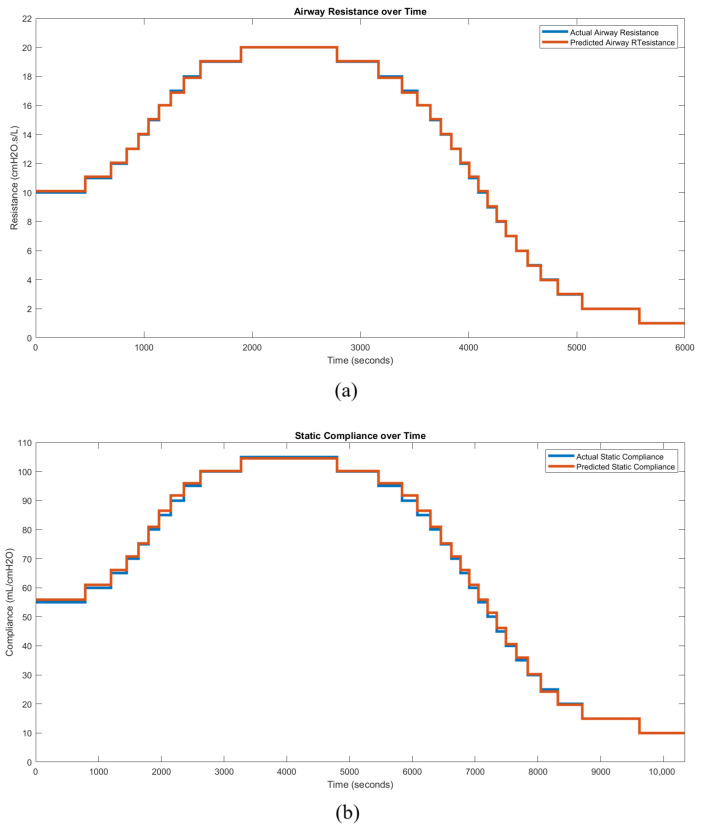
Condition trend monitoring predictions of PC mode for (**a**) $R_{RS}$ and (**b**) $C_{S}$.

**Table 1 diagnostics-14-02616-t001:** Accuracy and precision of extracted independent features.

Parameter	Residual Mean [%]	Residual STD [%]
PEEP	1.298	1.115
$Q_{max}$	−8.694	9.834
$V_{T}$	−5.406	3.941
$T_{i}$	−0.179	1.710
PIP	0.868	0.970

**Table 2 diagnostics-14-02616-t002:** Performance Summary of the Chosen Regression Models.

Case	Alg.	RMSE [%]	Training Speed [d]	Overfit Index [%]	Pred. Spd. [M Obs/s]	Size [MB]
R VCC	BLNN	1.610	2.421	3.742	0.61	25.042
R VCD	BLNN	1.390	3.386	0.781	0.94	25.043
R PC	WNN	0.431	0.632	1.176	0.33	23.127
C VCC	MNN	1.963	2.373	1.791	0.79	25.142
C VCD	NNN	2.297	2.331	0.323	1.10	25.142
C PC	MNN	0.924	0.028	5.967	0.54	23.208

**Table 3 diagnostics-14-02616-t003:** Local and total RMSE percentages per use case.

Use Case	Mean Local RMSE Percentage [%]	Mean Total RMSE Percentage [%]
R VCC	3.10	1.05
R VCD	3.44	1.24
R PC	0.72	0.26
C VCC	2.47	1.01
C VCD	6.04	1.78
C PC	1.34	0.77

**Table 4 diagnostics-14-02616-t004:** Interpreting the general performance metrics of the regression model clusters for this work.

Cluster	Training Time	Testing RMSE (Accuracy)	Model Size	Prediction Speed	Best For
Linear Models	Very Low	Moderate	Moderate to Large	Very High	Fast deployment with low computational cost
Tree-Based Models	Very Low	Moderate to High	Small to Moderate	High	Balanced performance, interpretability
Support Vector Machines	Moderate to High	Good to Very Good	Moderate	Low to Moderate	High accuracy with manageable training cost
Gaussian Process Regression	Very High	Very High (Excellent)	Large	Very Low	High-accuracy tasks where real-time prediction isn’t needed
Neural Networks	Moderate	Good	Compact	Very High	Real-time applications needing both accuracy and speed

## Data Availability

The simulation model, generated dataset, and statistical and machine learning models used in this study are available in a publicly accessible GitHub repository at: https://github.com/TheRealPieterMarx/Masters_2022.git (accessed on 17 November 2024).

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
