# Peer review of "A Technique for Monitoring Mechanically Ventilated Patient Lung Conditions"

_diagnostics, 2024, doi:10.3390/diagnostics14232616_

Round 1

Reviewer 1 Report

Comments and Suggestions for Authors

This manuscript provides an overview of a study on automated, non-invasive condition monitoring for mechanical ventilation, presenting background, methods, results, and conclusions. the manuscripts should be majorly revised  and then re-evaluate the work:

1.       Introduction is short and contribution is missing

2.       The manuscript presents a review of the available methods such as 2.1 to 2.5 which is difficult to follow

3.      Lack of Technical Depth in Methods: While the methods mention statistical classification and regression models and the use of ventilation waveform time-series data, the details of the used models are missing which makes them difficult to replicate

4.       There is no indication of the source or diversity of the data used to train the models, nor any mention of validation protocols.

5.       The authors do not discuss potential limitations or challenges in deploying the solution in real clinical settings. Mentioning any anticipated barriers (e.g., calibration across different ventilator brands etc. )

  1. The manuscript doesn’t clarify whether this classifier uses machine learning algorithms, statistical analysis, or rule-based logic. Identifying the specific approach, would help clarify the model’s complexity and potential adaptability.

7.       Absence of Validation or Performance Metrics: It’s unclear how well the classifier performs, as limited validation results or accuracy metrics are provided. I

8.       Comparison with related work is missing

Comments on the Quality of English Language

English presentation is weak

Author Response

Comment 1: Introduction is short and contribution is missing.
Response 1: Thank you for pointing this out. We agree with this comment. Therefore, we have added a paragraph in the introduction to more clearly illustrate the novelty and contribution of this study. The updates can be found in Lines 67–77.

Comment 2: The manuscript presents a review of the available methods (2.1–2.5), which is difficult to follow.
Response 2: Thank you for your valuable feedback. We have revised this section to improve its clarity, readability, and language presentation. The changes can be found in Lines 113–210.

Comment 3: Lack of Technical Depth in Methods: While the methods mention statistical classification and regression models and the use of ventilation waveform time-series data, the details of the used models are missing, which makes them difficult to replicate.
Response 3: We appreciate this observation. To address this, we have added detailed descriptions of the statistical mode classifier process (Lines 270–275), the implementation of KPDF (Lines 276–278), and specific paragraphs describing the steps taken to train the machine learning regression models (Lines 319–325).

Comment 4: There is no indication of the source or diversity of the data used to train the models, nor any mention of validation protocols.
Response 4: Thank you for pointing out the ambiguity. We have ensured that details on the dataset source and diversity, as well as validation protocols, are provided in Lines 213–238.

Comment 5: The authors do not discuss potential limitations or challenges in deploying the solution in real clinical settings. Mentioning any anticipated barriers (e.g., calibration across different ventilator brands, etc.).
Response 5: Thank you for this valuable suggestion. We have added a dedicated section titled "Work Limitations" (Section 6) to discuss potential challenges and limitations in deploying this solution, including barriers such as calibration across ventilator brands.

Comment 6: The manuscript doesn’t clarify whether this classifier uses machine learning algorithms, statistical analysis, or rule-based logic. Identifying the specific approach would help clarify the model’s complexity and potential adaptability.
Response 6: We appreciate this suggestion. We have more clearly labeled the different facets of our approach, distinguishing between statistical and machine learning techniques in Lines 240–249. Additionally, we added a visual aid (Figure 4) to further clarify these distinctions.

Comment 7: Absence of Validation or Performance Metrics: It’s unclear how well the classifier performs, as limited validation results or accuracy metrics are provided.
Response 7: Thank you for this comment. We have ensured that validation and performance metrics are provided throughout Section 4.3, including true positives, false positives, and overall accuracies per mode (Lines 342–438). Additionally, we provided residual mean percentages and standard deviations in Table 1 and boxplots summarizing the RMSE percentages, training speeds, and other metrics in Figure 17.

Comment 8: Comparison with related work is missing.
Response 8: We agree with this observation. To address this, we added a paragraph contextualizing the work relative to similar automation attempts. These additions can be found in Lines 35–66.

Comment 9: English presentation is weak.
Response 9: Thank you for your feedback. We have thoroughly revised the manuscript to improve grammar, spelling, and flow. These changes have been applied throughout the manuscript to ensure adherence to British English conventions and to enhance readability.

You can find the updated version of revised manuscript attached to this reply.

Reviewer 2 Report

Comments and Suggestions for Authors

This paper proposes a method for monitoring mechanically ventilated patient lung conditions. Overall, the paper is well-written and organized. I recommend the authors to address the following comments:

-          Since the proposed method is composed of already existing techniques, the work novelty should be demonstrated.

-          You may add 'lung' to the keywords.

-          It would be interesting to mention some literature works that have treated the same task automatically (section 1).

-          The advantages of considering a classification step before the regression process should be mentioned.

-          it is not mentioned why the authors have opted for the skewness and kurtosis as features after considering the dependent / independent features

-          The used features may be highly correlated, there is no guarantee about that!

-          in the discussion part, it would be interesting to see some interpretations about the difference in performance between the different models.

-          The work limitations should be mentioned and discussed

Author Response

Comment 1: Since the proposed method is composed of already existing techniques, the work novelty should be demonstrated.
Response 1: Thank you for this observation. We have added a paragraph to the introduction to more clearly illustrate the novelty and contribution of our work. These updates can be found in Lines 67–77.

Comment 2: You may add 'lung' to the keywords.
Response 2: Thank you for this suggestion. We have added the word 'lung' to the keywords to improve the discoverability of our manuscript. This update is reflected in Line 18.

Comment 3: It would be interesting to mention some literature works that have treated the same task automatically (Section 1).
Response 3: We agree with this comment. A paragraph has been added to Section 1 to contextualize our work relative to similar automation attempts. These additions can be found in Lines 35–66.

Comment 4: The advantages of considering a classification step before the regression process should be mentioned.
Response 4: Thank you for pointing this out. We have added an explanation of the rationale and advantages of including a classification step before regression. These changes can be found in Lines 236–245, and a supporting visual aid (Figure 4) has also been added.

Comment 5: It is not mentioned why the authors have opted for skewness and kurtosis as features after considering the dependent/independent features.
Response 5: We appreciate this suggestion. We have provided an explanation of why skewness and kurtosis were selected as features, addressing their importance in capturing data characteristics. These details are included in Lines 303–312.

Comment 6: The used features may be highly correlated; there is no guarantee about that!
Response 6: Thank you for this important remark. While we did not perform a formal correlation check, we have elaborated on the necessity of using the selected features and their role in the study. This is discussed in Lines 303–312.

Comment 7: In the discussion part, it would be interesting to see some interpretations about the difference in performance between the different models.
Response 7: We agree with this insightful suggestion. Interpretations regarding the differences in performance between the models have been added to the discussion section. Specifically, we modified Figure 17 and added Table 4 to provide a comparative perspective. These changes are reflected in Lines 490–501.

Comment 8: The work limitations should be mentioned and discussed.
Response 8: Thank you for highlighting this. We have added a dedicated section titled "Work Limitations" (Section 6) to discuss potential challenges and limitations in deploying the proposed solution.

You can find the updated version of revised manuscript attached to this reply.

Reviewer 3 Report

Comments and Suggestions for Authors

Dear authors,

thank you for the opportunity to read and review the manuscript.

The topic is of great interest.

General comments

Automated diagnostic techniques can be useful to estimate the mechanical lung parameters, thus ensuring not only physiopathological feedback on lung mechanics but also future application for mechanical ventilation settings and treatment of different lung disease associated with different lung mechanics parameter [Barahona J, Sahli Costabal F, Hurtado DE. Machine learning modeling of lung mechanics: Assessing the variability and propagation of uncertainty in respiratory-system compliance and airway resistance. Comput Methods Programs Biomed. 2024 Jan;243:107888. doi: 10.1016/j.cmpb.2023.107888. Epub 2023 Nov 4. PMID: 37948910.]

Specific comments

The paper is well written and interesting, featuring the application of ML to lung mechanics.

The paper analyzed simulation data on controlled mechanical ventilation, excluding controlled-assisted mechanical ventilation and thus patient-ventilator dyssynchrony. Further studies will be needed to determine the application of ML to patients and to analyze lung mechanics on different patients, thus ensuring a personalized treatment and ventilation setting.

The application of AI to mechanical ventilation could be the future of mechanical ventilation evaluation, avoiding invasive monitoring of compliance and resistance and analyzing automatically all the parameters related to the possibility of VILI (i.e. mechanical power [Gattinoni, L., Collino, F. & Camporota, L. Mechanical power: meaning, uses and limitations. Intensive Care Med 49, 465–467 (2023). https://doi.org/10.1007/s00134-023-06991-3]).

What do the authors think about heterogeneity of the characteristics between the lung (e.g. dependent versus non dependent lung zones)? Can the use of ML on a simulation system include this heterogeneity?

Limitation section could be added.

Author Response

Comment 1: What do the authors think about heterogeneity of the characteristics between the lungs (e.g., dependent versus non-dependent lung zones)? Can the use of ML on a simulation system include this heterogeneity?
Response 1: Thank you for this thoughtful question. We agree that heterogeneity in lung characteristics, such as differences between dependent and non-dependent lung zones, is an important consideration. While the current simulation system does not explicitly account for this heterogeneity, we have added a discussion in the future work section (Lines 490–501) to highlight the need for incorporating these variations in future studies.

Comment 2: Limitation section could be added.
Response 2: Thank you for this suggestion. A dedicated section titled "Work Limitations" (Section 6) has been added to address the key limitations of the current study and highlight areas for future research.

Thank you for the positive feedback and great suggestions. You can find the updated version of revised manuscript attached to this reply.

Round 2

Reviewer 1 Report

Comments and Suggestions for Authors

The authors have addressed my previous concerns

Reviewer 2 Report

Comments and Suggestions for Authors

accepted